# 1,2,4-Oxadiazole-Based Bio-Isosteres of Benzamides: Synthesis, Biological Activity and Toxicity to Zebrafish Embryo

**DOI:** 10.3390/ijms22052367

**Published:** 2021-02-27

**Authors:** Sen Yang, Chao-Li Ren, Tian-Yang Ma, Wen-Qian Zou, Li Dai, Xiao-Yu Tian, Xing-Hai Liu, Cheng-Xia Tan

**Affiliations:** College of Chemical Engineering, Zhejiang University of Technology, Hangzhou 310014, China; y17764528124@163.com (S.Y.); 18708589843@163.com (C.-L.R.); m15755187338@163.com (T.-Y.M.); zwqxyes@163.com (W.-Q.Z.); 15157357729@163.com (L.D.); t895745887@163.com (X.-Y.T.); xhliu@zjut.edu.cn (X.-H.L.)

**Keywords:** benzamides, biological activity, 1,2,4-oxadiazole, synthesis, toxicity

## Abstract

To discover new compounds with broad spectrum and high activity, we designed a series of novel benzamides containing 1,2,4-oxadiazole moiety by bioisosterism, and 28 benzamides derivatives with antifungal activity were synthesized. These compounds were evaluated against four fungi: *Botrytis cinereal*, *FusaHum graminearum*, *Marssonina mali*, and *Thanatephorus cucumeris*. The results indicated that most of the compounds displayed good fungicidal activities, especially against *Botrytis cinereal*. For example, **10a** (84.4%), **10d** (83.6%), **10e** (83.3%), **10f** (83.1%), **10i** (83.3%), and **10l** (83.6%) were better than pyraclostrobin (81.4%) at 100 mg/L. In addition, the acute toxicity of **10f** to zebrafish embryo was 20.58 mg/L, which was classified as a low-toxicity compound.

## 1. Introduction

Chemical pesticides play a vital role in solving food problems. However, as people’s environmental awareness gradually deepens, high-efficiency, low-toxicity, and environmentally friendly pesticides have become an inevitable trend in the creation of new pesticides [1,2,3]. There is no doubt that heterocyclic structures are important feature in synthetic pesticides for their high-efficiency, various biological activities, and diversity of possible substituents [4,5,6,7,8,9,10,11,12,13,14,15,16,17,18,19,20,21,22,23,24,25]. 1,2,4-Oxadiazole heterocycle, as an important kind of the five-membered oxygen-nitrogen heterocycle, has exhibited a wide range of biological activities in the field of pesticides, such as insecticidal [26,27,28], antifungal [29,30], and herbicidal activities [31]. It has often been introduced as a synergist into the structure of pesticides in order to improve the biological activities of the compounds. In addition, the 1,2,4-oxadiazole heterocycle, as a bio-isostere of the amide bond, has better hydrolytic and metabolic properties.

Diamide insecticides have attracted a lot of attention due to their novel mechanism of action, high efficiency, and low toxicity [32,33,34,35,36,37,38,39,40,41,42]. Following Bayer’s report on the first diamide insecticide flubendiamide, the use of chlorantraniliprole, cyantraniliprole, cyclaniliprole, (Figure 1) and other products was launched successively. However, while this type of insecticide showed excellent insecticidal effects, it gradually exposed potential risks to environmental non-target organisms [43,44]. Broflanilide is a meta-diamide insecticide developed by Mitsui Chemicals and co-developed with BASF SE. Because of its novel mechanism of action, this product was expected to become a blockbuster product.

In view of these facts mentioned above, broflanilide was employed as the lead compound in this study. According to the principle of bioisosterism [45,46], we searched for the amide group of broflanilide in the 1,2,4-oxadiazole ring, replaced the benzene ring with a pyridine structure containing a thioether derivative, and designed (Figure 2) and synthesized (Scheme 1) a series of novel benzamides substituted with 1,2,4-oxadiazole. These new compounds were confirmed by ^1^H NMR, ^13^C NMR, and HRMS, and their insecticidal activities, fungicidal activities, and toxicity test of zebrafish embryo were studied.

## 2. Results and Discussion

### 2.1. Synthesis of Target Compounds

The synthetic pathways to target compounds **9** and **10** are shown in Scheme 1. Intermediate **I** was prepared from 3,6-dichloropyridinecarboxylic acid **1** as a starting material via methylation, thioetherification, and hydrolysis reactions. During the thioetherification reaction, the newly prepared potassium ethanethiolate should be slowly added to the DMF solution of methyl 3,6-dichloropicolinate, and the temperature should be controlled at 0–5 °C to avoid the by-products (Scheme 2). As in our previous procedures [47], intermediate **Ⅱ** was easily obtained. Methyl 2-chloro-5-(5-(6-chloro-3-(ethylthio) pyridin-2-yl)-1,2,4-oxadiazol-3-yl) benzoate **7** was synthesized by cyclization reaction from imtermediate **Ⅱ** and 6-chloro-3-(ethylthio)picolinoyl chloride that had been synthesized from intermediate **Ⅰ**. Then, the compound **7** was hydrolyzed and spliced with substituted anilines to give a series of target compounds **9a**–**9n**. Finally, compound **9** was oxidized with mCPBA at room temperature to give target compound **10**, which avoided the impurities of pyridine-N-oxide. The characterization data for all synthesized compounds are provided in the supporting information file (Appendix A).

### 2.2. Spectrum Analysis of Target Compounds

All the target compounds were confirmed by ^1^H NMR, ^13^C NMR, and HRMS. In the ^1^H NMR spectra of **10a**, the -NH proton signals were found at δ 10.70 ppm. The signals of CH of the benzene and pyridine rings were assigned at 8.61–7.13 ppm. The signals at δ 3.78 ppm and 1.26 ppm were -CH_2_ and -CH_3_ peak, respectively. In the ^13^C-NMR spectra of compound **10a**, the appearances of signals at 166.81 and 163.79 ppm were assigned to the carbons of the 1,2,4-oxadiazole ring. Finally, all of the novel benzamides substituted with 1,2,4-oxadiazole exhibited a strong [M + H]^+^ peak in positive ion high-resolution electrospray mass spectra (HR-ESI-MS) analysis.

### 2.3. Biological Activities of Target Compounds

In Table 1, The target compounds had weak death rates against *Mythimna sepatara* (5–40%) at 500 mg/L, which were lower than the control drug broflanilide (100%). In addition, all of the target compounds had good fungicidal activities at 100 mg/L. Overall, the target compounds had better inhibitory activities against *Botrytis cinereal* than *Fusarium graminearum*, *Marssonina mali*, and *Thanatephorus cucumeris*. In particular, the inhibitory activity of compounds **10a** (84.4%), **10d** (83.6%), **10e** (83.3%), **10f** (83.1%), **10i** (83.3%), and **10l** (83.6%) were better than pyraclostrobin (81.4%), and **10b** (80.8%), **10g** (81.1%), **10h** (81.4%), and **10k** (81.9%) were comparable to pyraclostrobin. At the same time, compounds **9e** (57.7%), **9f** (63.5%), **9g** (65.9%), **9h** (57.1%), and **9k** (61.2%) also showed moderate activities. For *Fusarium graminearum* and *Marssonina mali*, compounds **9a**–**9n** displayed weak inhibition, both of which were less than 25%. Some of compound **10** had good activities (46.4–54.4%) but were inferior to pyraclostrobin. For *Thanatephorus cucumeris*, compound **9** had no inhibitory activity, and some of compound **10** exhibited moderate inhibitory activities (37.5–50.3%). From Table 2, we can see that compound **10f** had good inhibitory activity against *Botrytis cinereal* with EC_50_ of 14.44 μg/mL.

According to the insecticidal activity result, we speculated that the amide bridge bond played a key role in maintaining insecticidal activity, and it was likely that it would interact with the receptor through its hydrogen bond. Searching the amide bridge bond in 1,2,4-oxadiazole, we found that it lacked the corresponding hydrogen bond, and the group bulk increased, which resulted in the blocking of the binding of the compounds to the receptor and did not show a good insecticidal activity. It could be seen from Table 1 that the fungicidal activities of compound **10** was significantly higher than compound **9**, indicating that the structure containing ethylsulfonyl was beneficial in increasing the activity. The SAR of compound **10** in terms of fungicidal activities (Table 1) was that when there was no substituent on the benzene ring, the activity against *Botrytis cinereal* was superior to other compounds. In addition, by comparing the control efficacy of compounds **10k**, **10l**, **10m**, **10c**, and **10n**, we found that they showed that the para-position of aniline-containing substituents was not conducive to improving the activity.

### 2.4. Toxicity to Zebrafish Embryo

According to the fungicidal activity result, we selected compounds **9f** and **10f** with better activity to study the lethal and teratogenic effects exposure on zebrafish embryos from 6 to 96 hpf (Figure 3). When the **9f** concentration exceeded 2 mg/L, the mortality rate increased sharply. At 10 mg/L, the mortality rate reached as high as 90%. The resulting LC_50_ value for compound **9f** was 5.26 mg/L (Figure 3A). Similarly, the mortality rate of **10f** showed concentration-dependent curves (Figure 3) with a LC_50_ value of 20.58 mg/L (Figure 3B). Moreover, **9f** and **10f** produced similar teratogenic and decreased hatching effects on zebrafish embryos at 72 hpf (Figure 3C–F). At 72 hpf, the hatching rates of the compounds **9f** and **10f** under 10 mg/L exposure were about 43% and 82%, respectively. In addition, the malformation rate of **9f** was significantly higher than **10f** at the same concentration.

As the time and concentration increased, zebrafish embryos showed obvious developmental delay. At 76–96 hpf, a series of malformations appeared, such as delayed yolk absorption, pericardial cyst, lack of melanin, yolk sac, and bent spine (Figure 4). Among them, yolk cyst was the most obvious. By comparing the lethal and teratogenic effects of **9f** and **10f** exposure on zebrafish embryos, we were able to find that the toxicity of **9f** to zebrafish embryos was higher than that of **10f**. Thus, we speculated that the structure containing ethylsulfonyl was beneficial to reduce the toxicity to zebrafish embryos.

## 3. Experimental Section

### 3.1. General Information

Melting points were determined using an X-4 apparatus (Taike, Beijing, China) and the thermometer was uncorrected. ^1^H NMR and ^13^C NMR spectra were measured on BRUKER Avance 500 MHz spectrometer (Bruker 500 MHz, Fallanden, Switzerland) using CDCl_3_ or DMSO as the solvent. High-resolution electrospray mass spectra (HR-ESI–MS) were determined using an UPLC H CLASS/QTOF G2 XS mass spectrometer (Waters, Milford, CT, USA). All the reagents were analytical grade or synthesized in our laboratory.

Ethics statement: The Institutional Animal Care and Use Committee (IACUC) at Wenzhou Medical University (SYXK 2019-0009, April 4, 2019 to April 4, 2024) approved our study plan for proper use of zebrafish. All studies were carried out in strict accordance with the guidelines of the IACUC. All dissection was performed on ice, and all efforts were made to minimize suffering.

### 3.2. Synthesis

#### 3.2.1. Synthesis of Intermediate **Ⅰ**

Methyl 3,6-dichloropicolinate (**2**): To a stirred solution of 3,6-dichloropicolinic acid (**1**) (0.10 mmol) in acetone (10 mL), we added DMS (1.3 g, 0.01 mol) and K_2_CO_3_ (1.7 g). After stirring at 40 °C for 8 h, the mixture was cooled to room temperature and poured into water, the precipitation was filtered and dried to give 1.8 g light yellow solid. Yield: 90.1%, m.p. 54–55 °C ([48], 53–54 °C); ^1^H NMR (500 MHz, Chloroform-*d*) δ 7.77 (d, *J* = 8.5 Hz, 1H), 7.42 (d, *J* = 8.5 Hz, 1H), 4.00 (s, 3H).

Methyl 6-chloro-3-(ethylthio) picolinate (**3**): KTB (1.4 g), DMF (30 mL), and ethyl mercaptan (2.5 g, 0.04 mol) were added to a three-necked flask and reacted at 0 °C for about 1 h to give CH_3_CH_2_SK. The CH_3_CH_2_SK was dropped into a stirred solution of compound **2** (3.7 g) in DMF (20 mL) at 0 °C. After stirring at room temperature for 2 h, the mixture was quenched with water and extracted by EtOAc (100 mL). The extraction was dried over anhydrous MgSO_4_ and filtered. The filtration was concentrated and separated by column chromatography to give 3.1 g yellow solid. Yield: 74.5%, m.p. 128–129 °C; ^1^H NMR (500 MHz, Chloroform-*d*) δ 7.66 (d, *J* = 8.5 Hz, 1H), 7.41 (d, *J* = 8.5 Hz, 1H), 4.00 (s, 3H), 2.95 (q, *J* = 7.5 Hz, 2H), 1.39 (t, *J* = 7.5 Hz, 3H).

Intermediate **Ⅰ**: 30% NaOH (5 mL) was added to a solution of compound **3** (3.3 g) in THF. The mixture was then refluxed for 3 h. Afterwards, the mixture was cooled to room temperature and poured into water. Then, we adjusted the pH to 2–3, and 2.8 g white solid precipitate was obtained. Yield: 92.1%, m.p. 119–120 °C; ^1^H NMR (500 MHz, Chloroform-*d*) δ 10.76 (s, 1H), 7.71 (d, *J* = 8.5 Hz, 1H), 7.49 (d, *J* = 8.5 Hz, 1H), 2.97 (q, *J* = 7.5 Hz, 2H), 1.44 (t, 3H).

#### 3.2.2. Synthesis of Intermediate **Ⅱ**

The synthesis of intermediate **Ⅱ** refers to our previous work.

#### 3.2.3. Methyl 2-Chloro-5-(5-(6-Chloro-3-(Ethylthio) Pyridin-2-yl)-1,2,4-Oxadiazol-3-yl) Benzoate (**7**)

The intermediate **Ⅰ** (1.1 g, 5 mmol) and SOCl_2_ (20 mL) were added to a 100 mL flask and the mixture was refluxed for 6 h. Then, SOCl_2_ was removed under reduced pressure to give 6-chloro-3-(ethylthio)picolinoyl chloride.

To a solution of intermediate **Ⅱ** (1.1 g, 5.5 mmol) and triethylamine (1.2 g, 12 mmol) in toluene (100 mL), we added the prepared 6-chloro-3-(ethylthio)picolinoyl chloride dropwise at 0 °C for 1 h. The mixture was then refluxed for 2 h. Afterwards, the mixture was cooled to room temperature and washed by saturated sodium chloride solution (100 mL × 3). The organic layer was dried by Na_2_SO_4_ and removed to give yellow solid (1.4 g). Yield: 62.2%, m.p. 155–157 °C; ^1^H NMR (500 MHz, Chloroform-*d*) δ 8.63 (d, *J* = 2.0 Hz, 1H), 8.35–8.20 (m, 1H), 7.89 (d, *J* = 8.5 Hz, 1H), 7.66 (d, *J* = 8.0 Hz, 1H), 7.54 (d, *J* = 8.5 Hz, 1H), 3.87 (s, 3H), 3.17 (q, J = 7.0 Hz, 2H), 1.32 (t, J = 7.5 Hz, 3H); HRMS calcd for C_17_H_14_Cl_2_N_3_O_3_S [M + H]^+^ 410.0127, found 410.0126.

#### 3.2.4. 2-Chloro-5-(5-(6-Chloro-3-(Ethylthio) Pyridin-2-yl)-1,2,4-Oxadiazol-3-yl) Benzoic Acid (**8**)

We added 30% NaOH (5 mL) to a solution of compound **7** (0.8 g, 2.0 mmol) in THF. The mixture was then refluxed for 2 h. Afterwards, the mixture was cooled to room temperature and the solvent was removed. Then, we adjusted the pH to 2–3, and white solid precipitate was obtained (0.7 g). Yield: 90.8%, m.p. 225–227 °C; ^1^H NMR (500 MHz, Chloroform-*d*) δ 13.47 (s, 1H), 8.55 (d, *J* = 2.0 Hz, 1H), 8.36 (d, *J* = 8.5 Hz, 1H), 8.26–8.19 (m, 1H), 7.91 (d, *J* = 8.5 Hz, 1H), 7.80 (d, *J* = 8.0 Hz, 1H), 3.16 (q, J = 7.5 Hz, 2H), 1.30 (t, J = 7.0 Hz, 3H); HRMS calcd for C_16_H_12_Cl_2_N_3_O_3_S [M + H]^+^ 395.9971, found 395.9973.

#### 3.2.5. Synthesis of Target Compound **9**

To a solution of compound **8** (1.0 g, 2.0 mmol), EDCI (0.1 g) and triethylamine (0.2 g) in DCM (100 mL), we added the substituted aniline (3.0 mmol) at 0 °C, and the mixture was stirred for 8 h to give compound **9** by the method of column chromatography separation.

*2-Chloro-5-(5-(6-chloro-3-(ethylthio)pyridin-2-yl)-1,2,4-oxadiazol-3-yl)-N-phenylbenzamide* (***9a***): White solid, yield 73.4%, m.p. 243–245 °C; ^1^H NMR (500 MHz, DMSO-*d*_6_) ^1^H NMR (500 MHz, Chloroform-*d*) δ 10.70 (s, 1H), 8.23–8.18 (m, 2H), 8.13 (d, *J* = 9.0 Hz, 1H), 7.88–7.84 (m, 1H), 7.79 (d, *J* = 8.5 Hz, 1H), 7.73 (d, *J* = 8.0 Hz, 2H), 7.38 (t, *J* = 8.0 Hz, 2H), 7.14 (t, *J* = 7.0 Hz, 1H), 3.15 (q, *J* = 7.5 Hz, 2H), 1.30 (t, *J* = 7.0 Hz, 3H); ^13^C NMR (500 MHz, DMSO-*d*_6_) δ 172.43, 167.11, 164.10, 145.88, 138.71, 138.64, 138.22, 138.13, 137.88, 135.87, 133.66, 131.22, 129.60, 129.01, 127.81, 127.29, 124.30, 119.89, 25.53, 13.36; HRMS calcd for C_22_H_17_Cl_2_N_4_O_2_S [M + H]^+^ 471.0444, found 471.0447.

*2-Chloro-5-(5-(6-chloro-3-(ethylthio)pyridin-2-yl)-1,2,4-oxadiazol-3-yl)-N-(o-tolyl)benzamide* (***9b***): White solid, yield 77.4%, m.p. 230–232 °C; ^1^H NMR (500 MHz, DMSO-*d*_6_) δ 10.19 (s, 1H), 8.26 (d, *J* = 2.0 Hz, 1H), 8.20 (dd, *J* = 8.5, 2.0 Hz, 1H), 8.14 (d, *J* = 9.0 Hz, 1H), 7.86 (d, *J* = 8.5 Hz, 1H), 7.80 (d, *J* = 8.5 Hz, 1H), 7.49 (d, *J* = 7.5 Hz, 1H), 7.29 (d, *J* = 7.5 Hz, 1H), 7.25 (t, *J* = 6.5 Hz, 1H), 7.19 (t, *J* = 7.5 Hz, 1H), 3.17 (q, *J* = 7.0 Hz, 2H), 2.30 (s, 3H), 1.32 (t, *J* = 7.5 Hz, 3H); ^13^C NMR (500 MHz, DMSO-*d*_6_) ^13^C NMR (126 MHz, DMSO-*d*_6_) δ 172.45, 166.07, 163.92, 145.90, 138.67, 137.96, 137.61, 136.24, 133.69, 133.53, 133.38, 132.91, 131.23, 130.94, 129.56, 129.38, 127.83, 127.29, 125.25, 119.91, 25.55, 20.64, 13.38; HRMS calcd for C_23_H_19_Cl_2_N_4_O_2_S [M + H]^+^ 485.0600, found 485.0600.

*2-Chloro-5-(5-(6-chloro-3-(ethylthio)pyridin-2-yl)-1,2,4-oxadiazol-3-yl)-N-(p-tolyl)benzamide* (***9c***): White solid, yield 74.3%, m.p. 251–253 °C; ^1^H NMR (500 MHz, DMSO-*d*_6_) δ 10.61 (s, 1H), 8.19 (m, 2H), 8.13 (d, *J* = 9.0 Hz, 1H), 7.85 (d, *J* = 8.5 Hz, 1H), 7.79 (d, *J* = 9.0 Hz, 1H), 7.61 (d, *J* = 8.0 Hz, 2H), 7.18 (d, *J* = 8.0 Hz, 2H), 3.16 (d, *J* = 7.5 Hz, 2H), 2.29 (s, 3H), 1.30 (t, *J* = 7.5 Hz, 3H); ^13^C NMR (500 MHz, DMSO-*d*_6_) δ 172.46, 167.15, 163.96, 145.90, 138.63, 138.29, 137.98, 137.57, 136.26, 133.71, 133.41, 131.25, 129.58, 129.41, 127.85, 127.32, 125.02, 119.93, 25.56, 20.67, 13.39; HRMS calcd for C_23_H_19_Cl_2_N_4_O_2_S [M + H]^+^ 485.0600, found 485.0606.

*N-(4-(tert-Butyl) phenyl)-2-chloro-5-(5-(6-chloro-3-(ethylthio) pyridin-2-yl)-1,2,4-oxadiazol-3-yl) benzamide* (***9d***): White solid, yield 78.4%,. m.p. 243–244 °C; ^1^H NMR (500 MHz, DMSO-*d*_6_) δ 10.63 (s, 1H), 8.23–8.17 (m, 2H), 8.14 (d, *J* = 9.0 Hz, 1H), 7.85 (d, *J* = 8.0 Hz, 1H), 7.79 (d, *J* = 8.5 Hz, 1H), 7.64 (d, *J* = 8.5 Hz, 2H), 7.39 (d, *J* = 8.5 Hz, 2H), 3.16 (q, *J* = 7.5 Hz, 2H), 1.34–1.26 (m, 12H); ^13^C NMR (500 MHz, DMSO-*d*_6_) δ 172.46, 167.40, 167.19, 138.54, 138.11, 135.64, 133.72, 133.30, 133.04, 133.00, 131.21, 127.88, 127.53, 126.84, 126.17, 124.98, 25.64, 20.70, 18.01, 13.34; HRMS calcd for C_26_H_25_Cl_2_N_4_O_2_S [M + H]^+^ 527.1070, found 527.1071.

*2-Chloro-5-(5-(6-chloro-3-(ethylthio)pyridin-2-yl)-1,2,4-oxadiazol-3-yl)-N-(2,4-dimethylphenyl)benzamide* (***9e***): Grey solid, yield 69.2%, m.p. 255–256 °C; ^1^H NMR (500 MHz, DMSO-*d*_6_) δ 10.10 (s, 1H), 8.24 (d, *J* = 2.0 Hz, 1H), 8.19 (dd, *J* = 8.0, 2.0 Hz, 1H), 8.14 (d, *J* = 8.5 Hz, 1H), 7.83 (dd, *J* = 24.0, 8.5 Hz, 2H), 7.34 (d, *J* = 8.0 Hz, 1H), 7.10 (s, 1H), 7.05 (d, *J* = 8.0 Hz, 1H), 3.17 (q, *J* = 7.0 Hz, 2H), 2.28 (d, *J* = 6.0 Hz, 6H), 1.32 (t, *J* = 7.5 Hz, 3H); ^13^C NMR (500 MHz, DMSO-*d*_6_) δ 172.46, 167.40, 167.19, 138.54, 138.11, 135.64, 133.72, 133.30, 133.04, 131.21, 127.88, 127.53, 126.84, 126.17, 25.64, 20.70, 18.01, 13.34; HRMS calcd for C_24_H_21_Cl_2_N_4_O_2_S [M + H]^+^ 499.0757, found 499.0763.

*2-Chloro-5-(5-(6-chloro-3-(ethylthio)pyridin-2-yl)-1,2,4-oxadiazol-3-yl)-N-(3-(trifluoromethyl)phenyl)benzamide* (***9f***): White solid, yield 77.8%, m.p. 211–214 °C; ^1^H NMR (500 MHz, DMSO-*d*_6_) δ 11.06 (s, 1H), 8.27 (d, *J* = 2.0 Hz, 1H), 8.25–8.21 (m, 2H), 8.14 (d, *J* = 9.0 Hz, 1H), 7.92 (d, *J* = 8.0 Hz, 1H), 7.89 (d, *J* = 8.5 Hz, 1H), 7.80 (d, *J* = 9.0 Hz, 1H), 7.64 (t, *J* = 8.0 Hz, 1H), 7.51 (d, *J* = 8.0 Hz, 1H), 3.16 (q, *J* = 7.5 Hz, 2H), 1.30 (t, *J* = 7.0 Hz, 3H); ^13^C NMR (500 MHz, DMSO-*d*_6_) δ 172.48, 167.10, 164.59, 145.91, 139.49, 138.68, 138.63, 138.59, 138.28, 137.58, 137.35, 133.70, 131.35, 131.29, 130.37, 127.88, 127.81, 125.11, 123.53, 120.68, 116.00, 25.57, 13.36; HRMS calcd for C_23_H_16_Cl_2_F_3_N_4_O_2_S [M + H]^+^ 539.0318, found 539.0322.

*2-Chloro-N-(3-chloro-2-methylphenyl)-5-(5-(6-chloro-3-(ethylthio) pyridin-2-yl)-1,2,4-oxadiazol-3-yl) benzamide* (***9g***): Yellow solid, yield 79.7%. m.p. 225–226 °C; ^1^H NMR (500 MHz, DMSO-*d*_6_) δ 10.44 (s, 1H), 8.29 (d, *J* = 1.5 Hz, 1H), 8.21 (dd, *J* = 8.0, 1.5 Hz, 1H), 8.14 (d, *J* = 9.0 Hz, 1H), 7.87 (d, *J* = 8.0 Hz, 1H), 7.80 (d, *J* = 9.0 Hz, 1H), 7.47 (d, *J* = 8.0 Hz, 1H), 7.39 (d, *J* = 8.0 Hz, 1H), 7.29 (t, *J* = 8.0 Hz, 1H), 3.16 (q, *J* = 7.5 Hz, 2H), 2.35 (s, 3H), 1.32 (t, *J* = 7.5 Hz, 3H); ^13^C NMR (500 MHz, DMSO-*d*_6_) δ 172.39, 167.05, 164.53, 145.85, 138.65, 138.24, 137.61, 137.10, 134.00, 133.62, 131.53, 131.19, 129.61, 127.77, 127.40, 127.18, 127.14, 127.08, 125.45, 124.97, 25.54, 15.30, 13.26; HRMS calcd for C_23_H_18_Cl_3_N_4_O_2_S [M + H]^+^ 519.0211, found 519.0211.

*2-Chloro-5-(5-(6-chloro-3-(ethylthio)pyridin-2-yl)-1,2,4-oxadiazol-3-yl)-N-(2-fluorophenyl)benzamide* (***9h***): White solid, yield 66.5%, m.p. 213–217 °C; ^1^H NMR (500 MHz, DMSO-*d*_6_) ^1^H NMR (500 MHz, Chloroform-*d*) δ 10.56 (s, 1H), 8.21 (m, 2H), 8.14 (d, *J* = 8.5 Hz, 1H), 7.91–7.82 (m, 2H), 7.80 (d, *J* = 8.5 Hz, 1H), 7.36–7.21 (m, 3H), 3.16 (q, *J* = 7.5 Hz, 2H), 1.31 (t, *J* = 7.0 Hz, 3H); ^13^C NMR (500 MHz, DMSO-*d*_6_) ^13^C NMR (126 MHz, DMSO-*d*_6_) δ 172.42, 164.54, 158.63, 145.88, 143.26, 139.54, 138.66, 138.23, 137.37, 136.91, 132.59, 131.18, 129.68, 127.80, 127.49, 125.78, 125.19, 124.60 (d, *J* = 10 Hz), 123.80, 25.55, 13.31; HRMS calcd for C_22_H_16_Cl_2_FN_4_O_2_S [M + H]^+^ 489.0350, found 489.0355.

*2-Chloro-5-(5-(6-chloro-3-(ethylthio)pyridin-2-yl)-1,2,4-oxadiazol-3-yl)-N-(4-fluorophenyl)benzamide* (***9i***): Yellow solid, yield 61.6%, m.p. 268–270 °C; ^1^H NMR (500 MHz, DMSO-*d*_6_) δ 10.76 (s, 1H), 8.24–8.18 (m, 2H), 8.13 (d, *J* = 9.0 Hz, 1H), 7.89–7.83 (m, 1H), 7.82–7.72 (m, 3H), 7.22 (t, *J* = 9.0 Hz, 2H), 3.16 (q, *J* = 7.5 Hz, 2H), 1.30 (t, *J* = 7.5 Hz, 3H); ^13^C NMR (500 MHz, DMSO-*d*_6_) δ 172.41, 167.06, 163.96, 145.85, 138.61, 138.19, 137.70, 137.57, 133.64, 131.20, 129.64, 127.77, 127.28, 124.99, 121.74, 121.68, 115.67, 115.49, 25.50, 13.33; HRMS calcd for C_22_H_16_Cl_2_FN_4_O_2_S [M + H]^+^ 489.0350, found 489.0350.

*2-Chloro-5-(5-(6-chloro-3-(ethylthio)pyridin-2-yl)-1,2,4-oxadiazol-3-yl)-N-(2,6-difluorophenyl)benzamide* (***9j***): Grey solid, yield 57.8%, m.p. 258–261 °C; ^1^H NMR (500 MHz, DMSO-*d*_6_) δ 10.57 (s, 1H), 8.27–8.18 (m, 2H), 8.14 (d, *J* = 8.5 Hz, 1H), 7.90–7.76 (m, 3H), 7.39 (t, *J* = 8.5 Hz, 1H), 7.16 (t, *J* = 8.5 Hz, 1H), 3.16 (q, *J* = 7.5 Hz, 2H), 1.31 (t, *J* = 7.5 Hz, 3H); ^13^C NMR (500 MHz, DMSO-*d*_6_) δ 172.46, 167.78, 167.11, 158.28, 152.27, 146.15, 145.92, 138.69, 138.25, 137.62, 137.22, 133.78, 131.25, 129.81, 127.84, 127.51, 117.00, 116.01, 111.58, 25.59, 13.33; HRMS calcd for C_22_H_15_Cl_2_F_2_N_4_O_2_S [M + H]^+^ 507.0255, found 507.0258.

*2-Chloro-5-(5-(6-chloro-3-(ethylthio)pyridin-2-yl)-1,2,4-oxadiazol-3-yl)-N-(2-chlorophenyl)benzamide* (***9k***): White solid, yield 75.0%, m.p. 207–208 °C; ^1^H NMR (500 MHz, DMSO-*d*_6_) δ 10.48 (s, 1H), 8.30 (s, 1H), 8.21 (d, *J* = 8.5 Hz, 1H), 8.14 (d, *J* = 8.5 Hz, 1H), 7.86 (d, *J* = 8.5 Hz, 1H), 7.80 (d, *J* = 8.5 Hz, 1H), 7.73 (d, *J* = 7.5 Hz, 1H), 7.58 (d, *J* = 8.0 Hz, 1H), 7.42 (t, *J* = 7.0 Hz, 1H), 7.32 (t, *J* = 7.5 Hz, 1H), 3.16 (q, *J* = 7.0 Hz, 2H), 1.32 (t, *J* = 7.5 Hz, 3H); ^13^C NMR (500 MHz, DMSO-*d*_6_) δ 172.45, 167.07, 164.37, 145.87, 140.12, 138.25, 137.55, 137.44, 133.65, 133.32, 131.24, 130.76, 129.93, 129.85, 127.78, 127.29, 125.06, 119.40, 119.32, 118.33, 25.54, 13.35; HRMS calcd for C_22_H_16_Cl_3_N_4_O_2_S [M + H]^+^ 505.0054, found 505.0055.

*2-Chloro-5-(5-(6-chloro-3-(ethylthio)pyridin-2-yl)-1,2,4-oxadiazol-3-yl)-N-(3-chlorophenyl)benzamide* (***9l***): White solid, yield 75.4%, m.p. 232–234 °C; ^1^H NMR (500 MHz, DMSO-*d*_6_) δ 10.90 (s, 1H), 8.26–8.20 (m, 2H), 8.14 (d, *J* = 8.5 Hz, 1H), 7.94 (t, *J* = 1.5 Hz, 1H), 7.87 (d, *J* = 8.5 Hz, 1H), 7.80 (d, *J* = 9.0 Hz, 1H), 7.60 (d, *J* = 8.0 Hz, 1H), 7.42 (t, *J* = 8.0 Hz, 1H), 7.24–7.18 (m, 1H), 3.16 (q, *J* = 7.0 Hz, 2H), 1.30 (t, *J* = 7.0 Hz, 3H); ^13^C NMR (126 MHz, DMSO-*d*_6_) ^13^C NMR (126 MHz, Chloroform-*d*) δ 177.49, 172.13, 169.39, 150.90, 145.27, 143.38, 142.55, 138.73, 138.41, 136.35, 136.28, 135.80, 134.93, 132.82, 132.51, 130.17, 129.06, 124.45, 124.38, 123.35, 30.60, 18.41; HRMS calcd for C_22_H_16_Cl_3_N_4_O_2_S [M + H]^+^ 505.0054, found 505.0055.

*2-Chloro-5-(5-(6-chloro-3-(ethylthio)pyridin-2-yl)-1,2,4-oxadiazol-3-yl)-N-(4-chlorophenyl)benzamide* (***9m***): White solid, yield 77.3%, m.p. 242–243 °C; ^1^H NMR (500 MHz, DMSO-*d*_6_) δ 10.84 (s, 1H), 8.24–8.18 (m, 2H), 8.11 (d, *J* = 9.0 Hz, 1H), 7.85 (d, *J* = 8.0 Hz, 1H), 7.77 (m, 3H), 7.43 (d, *J* = 9.0 Hz, 2H), 3.14 (q, *J* = 7.5 Hz, 2H), 1.29 (t, *J* = 7.5 Hz, 3H); ^13^C NMR (126 MHz, DMSO-*d*_6_) δ 172.42, 167.07, 164.17, 145.86, 138.55, 138.25, 137.66, 137.57, 137.52, 133.65, 131.22, 129.75, 128.92, 127.92, 127.78, 127.33, 125.04, 121.45, 25.54, 13.33; HRMS calcd for C_22_H_16_Cl_3_N_4_O_2_S [M + H]^+^ 505.0054, found 505.0055.

*N-(4-Bromophenyl)-2-chloro-5-(5-(6-chloro-3-(ethylthio)pyridin-2-yl)-1,2,4-oxadiazol-3-yl)benzamide* (***9n***): Yellow solid, yield 78.8%, m.p. 267–269 °C; ^1^H NMR (500 MHz, DMSO-*d*_6_) δ 10.84 (s, 1H), 8.25–8.19 (m, 2H), 8.13 (d, *J* = 8.5 Hz, 1H), 7.87 (d, *J* = 8.0 Hz, 1H), 7.79 (d, *J* = 9.0 Hz, 1H), 7.72 (d, *J* = 9. Hz, 2H), 7.57 (d, *J* = 8.5 Hz, 2H), 3.16 (q, *J* = 7.5 Hz, 2H), 1.30 (t, *J* = 7.0 Hz, 3H); ^13^C NMR (126 MHz, DMSO-*d*_6_) δ 172.43, 167.08, 164.20, 145.88, 138.60, 138.24, 138.08, 137.57, 133.64, 131.84, 131.24, 129.77, 127.80, 127.33, 125.04, 121.83, 115.99, 25.55, 13.35; HRMS calcd for C_22_H_16_BrCl_2_N_4_O_2_S [M + H]^+^ 548.9549, found 548.9554.

#### 3.2.6. Synthesis of Target Compound **10**

To a stirred solution of compound **9** (1.0 mmol) in DCM (20 mL), we added mCPBA (0.5 g, 3.0 mmol). After stirring at room temperature for 3 h, the mixture was poured into water and the pH was adjusted to 7–8 with NaHCO_3_. The organic layer was dried by Na_2_SO_4_ and removed under reduced pressure to give compound **10**.

*2-Chloro-5-(5-(6-chloro-3-(ethylsulfonyl)pyridin-2-yl)-1,2,4-oxadiazol-3-yl)-N-phenylbenzamide* (***10a***): Pink solid, yield 83.7%, m.p. 263–267 °C; ^1^H NMR (500 MHz, DMSO-*d*_6_) δ 10.70 (s, 1H), 8.61 (d, *J* = 8.5 Hz, 1H), 8.23 (d, *J* = 2.5 Hz, 1H), 8.20 (m, 2H), 7.85 (d, *J* = 8.5 Hz, 1H), 7.74 (d, *J* = 8.0 Hz, 2H), 7.37 (t, *J* = 8.0 Hz, 2H), 7.13 (t, *J* = 7.5 Hz, 1H), 3.78 (q, *J* = 7.5 Hz, 2H), 1.26 (t, *J* = 7.5 Hz, 3H); ^13^C NMR (500 MHz, DMSO-*d*_6_) δ 171.62, 166.81, 163.79, 154.63, 142.95, 142.22, 138.64, 137.80, 135.62, 133.90, 131.16, 129.50, 128.82, 127.36, 124.27, 124.07, 119.71, 50.37, 6.64; HRMS calcd for C_22_H_17_Cl_2_N_4_O_4_S [M + H]^+^ 503.0342, found 503.0347.

*2-Chloro-5-(5-(6-chloro-3-(ethylsulfonyl)pyridin-2-yl)-1,2,4-oxadiazol-3-yl)-N-(o-tolyl)benzamide* (***10b***): White solid, yield 77.4%, m.p. 257–259 °C; ^1^H NMR (500 MHz, DMSO-*d*_6_) δ 10.18 (s, 1H), 8.62 (d, *J* = 9.0 Hz, 1H), 8.26 (d, *J* = 2.0 Hz, 1H), 8.20 (d, *J* = 8.5 Hz, 2H), 7.85 (d, *J* = 8.5 Hz, 1H), 7.49 (d, *J* = 7.5 Hz, 1H), 7.31–7.20 (m, 2H), 7.18 (td, *J* = 7.5, 1.5 Hz, 1H), 3.79 (q, *J* = 7.5 Hz, 2H), 2.31 (s, 3H), 1.27 (t, *J* = 7.5 Hz, 3H); ^13^C NMR (500 MHz, DMSO-*d*_6_) δ 171.64, 166.85, 164.14, 154.64, 142.98, 142.23, 137.97, 135.64, 135.47, 133.89, 132.94, 131.14, 130.61, 130.44, 129.37, 128.79, 127.38, 126.07, 124.24, 50.37, 39.93, 17.90, 6.63; HRMS calcd for C_23_H_19_Cl_2_N_4_O_4_S [M + H]^+^ 517.0499, found 517.0500.

*2-Chloro-5-(5-(6-chloro-3-(ethylsulfonyl)pyridin-2-yl)-1,2,4-oxadiazol-3-yl)-N-(p-tolyl)benzamide* (***10c***): White solid, yield 84.3%, m.p. 260–261 °C; ^1^H NMR (500 MHz, DMSO-*d*_6_) δ 10.59 (s, 1H), 8.61 (d, *J* = 8.5 Hz, 1H), 8.26–8.15 (m, 3H), 7.84 (d, *J* = 8.5 Hz, 1H), 7.61 (d, *J* = 8.5 Hz, 2H), 7.17 (d, *J* = 8.0 Hz, 2H), 3.77 (q, *J* = 7.5 Hz, 2H), 2.28 (s, 3H), 1.26 (t, *J* = 7.0 Hz, 3H); ^13^C NMR (500 MHz, DMSO-*d*_6_) ^13^C NMR (126 MHz, DMSO-*d*_6_) δ 171.61, 166.81, 163.60, 154.63, 142.94, 142.20, 137.85, 136.11, 135.60, 133.90, 133.14, 131.16, 129.45, 129.18, 128.78, 127.32, 124.22, 119.72, 50.36, 20.46, 6.61; HRMS calcd for C_23_H_19_Cl_2_N_4_O_4_S [M + H]^+^ 517.0499, found 517.0505.

*N-(4-(tert-Butyl)phenyl)-2-chloro-5-(5-(6-chloro-3-(ethylsulfonyl)pyridin-2-yl)-1,2,4-oxadiazol-3-yl)benzamide* (***10d***): Grey yield, yield 79.2%, m.p. 268–270 °C; ^1^H NMR (500 MHz, DMSO-*d*_6_) δ 10.61 (s, 1H), 8.62 (d, *J* = 8.5 Hz, 1H), 8.24–8.12 (m, 3H), 7.85 (d, *J* = 9.0 Hz, 1H), 7.64 (d, *J* = 8.5 Hz, 2H), 7.39 (d, *J* = 8.5 Hz, 2H), 3.78 (q, *J* = 7.5 Hz, 2H), 1.33–1.27 (m, 12H); ^13^C NMR (500 MHz, DMSO-*d*_6_) δ 176.85, 172.05, 168.89, 159.88, 151.78, 148.21, 147.45, 143.08, 141.26, 140.83, 139.14, 136.39, 134.64, 134.06, 132.50, 130.66, 129.46, 124.77, 55.61, 39.28, 36.36, 11.87; HRMS calcd for C_26_H_25_Cl_2_N_4_O_4_S [M + H]^+^ 559.0968, found 559.0971.

*2-Chloro-5-(5-(6-chloro-3-(ethylsulfonyl)pyridin-2-yl)-1,2,4-oxadiazol-3-yl)-N-(2,4-dimethylphenyl)benzamide* (***10e***): White solid, yield 78.4%,. m.p. 246–248 °C; ^1^H NMR (500 MHz, DMSO-*d*_6_) δ 10.09 (s, 1H), 8.61 (d, *J* = 8.5 Hz, 1H), 8.24 (d, *J* = 2.0 Hz, 1H), 8.22–8.17 (m, 2H), 7.84 (d, *J* = 8.5 Hz, 1H), 7.34 (d, *J* = 8.0 Hz, 1H), 7.08 (s, 1H), 7.04 (dd, *J* = 8.0, 2.0 Hz, 1H), 3.79 (q, *J* = 7.0 Hz, 2H), 2.27 (d, *J* = 7.5 Hz, 6H), 1.27 (t, *J* = 7.5 Hz, 3H); ^13^C NMR (500 MHz, DMSO-*d*_6_) δ 171.62, 166.84, 164.16, 154.64, 142.96, 142.21, 138.02, 135.62, 135.34, 133.87, 132.84, 132.81, 131.12, 130.92, 129.32, 128.78, 127.34, 126.57, 125.93, 124.20, 50.37, 20.49, 17.81, 6.62; HRMS calcd for C_24_H_21_Cl_2_N_4_O_4_S [M + H]^+^ 531.0655, found 531.0660.

*2-Chloro-5-(5-(6-chloro-3-(ethylsulfonyl)pyridin-2-yl)-1,2,4-oxadiazol-3-yl)-N-(3-(trifluoromethyl)phenyl)benzamide* (***10f***): Yellow solid, yield 76.8%, m.p. 214–217 °C; H NMR (500 MHz, DMSO-*d*_6_) δ 11.07 (s, 1H), 8.61 (d, *J* = 8.5 Hz, 1H), 8.30 (d, *J* = 2.0 Hz, 1H), 8.26–8.17 (m, 3H), 7.92 (d, *J* = 8.5 Hz, 1H), 7.87 (d, *J* = 8.5 Hz, 1H), 7.62 (t, *J* = 8.0 Hz, 1H), 7.50 (d, *J* = 7.5 Hz, 1H), 3.78 (q, *J* = 7.5 Hz, 2H), 1.26 (t, *J* = 7.0 Hz, 3H); ^13^C NMR (500 MHz, DMSO-*d*_6_) δ 171.64, 166.77, 164.23, 154.62, 142.95, 142.20, 139.37, 137.23, 135.61, 133.90, 131.23, 130.13, 129.80, 129.54 (d, *J* = 32.1 Hz), 128.79, 127.49, 125.10, 124.32, 123.30, 120.44 (d), 115.78 (d, *J* = 4.0 Hz), 50.37, 6.62; HRMS calcd for C_23_H_16_Cl_2_F_3_N_4_O_4_S [M + H]^+^ 571.0216, found 571.0222.

2*-Chloro-N-(3-chloro-2-methylphenyl)-5-(5-(6-chloro-3-(ethylsulfonyl)pyridin-2-yl)-1,2,4-oxadiazol-3-yl)benzamide* (***10g***): Yellow solid, yield 69.1%, m.p. 255–256 °C; ^1^H NMR (500 MHz, DMSO-*d*_6_) δ 10.44 (s, 1H), 8.62 (d, *J* = 8.5 Hz, 1H), 8.29 (d, *J* = 2.0 Hz, 2H), 8.21 (m, 1H), 7.86 (d, *J* = 8.5 Hz, 1H), 7.46 (d, *J* = 8.0 Hz, 1H), 7.38 (d, *J* = 7.5 Hz, 1H), 7.28 (t, *J* = 8.0 Hz, 1H), 3.79 (q, *J* = 7.0 Hz, 2H), 2.34 (s, 3H), 1.27 (t, *J* = 7.5 Hz, 3H); ^13^C NMR (500 MHz, DMSO-*d*_6_) ^13^C NMR (126 MHz, DMSO-*d*_6_) δ 171.64, 166.81, 164.30, 154.63, 142.97, 142.21, 137.60, 137.00, 135.63, 133.88, 131.45, 131.17, 129.55, 128.81, 128.78, 127.40, 127.01, 126.95, 125.34, 124.28, 50.36, 15.19, 6.64; HRMS calcd for C_23_H_18_Cl_3_N_4_O_4_S [M + H]^+^ 551.0109, found 551.0108.

*2-Chloro-5-(5-(6-chloro-3-(ethylsulfonyl)pyridin-2-yl)-1,2,4-oxadiazol-3-yl)-N-(2-fluorophenyl)benzamide* (***10h***): Brown solid, yield 73.7%, m.p. 253–257 °C; ^1^H NMR (500 MHz, DMSO-*d*_6_) δ 10.58 (s, 1H), 8.61 (d, *J* = 8.5 Hz, 1H), 8.27–8.11 (m, 3H), 7.94–7.81 (m, 2H), 7.38–7.17 (m, 3H), 3.78 (q, *J* = 7.0 Hz, 2H), 1.26 (t, *J* = 7.0 Hz, 3H); ^13^C NMR (500 MHz, DMSO-*d*_6_) ^13^C NMR (126 MHz, DMSO-*d*_6_) δ 171.62, 166.80, 166.07, 154.64, 142.94, 142.19, 137.26, 135.60, 133.97, 133.31, 132.69, 131.16, 130.64, 129.60, 128.77, 127.87, 127.51, 125.62, 124.45, 115.80 (d, *J* = 77.5 Hz), 50.37, 6.58; HRMS calcd for C_22_H_16_Cl_2_FN_4_O_4_S [M + H]^+^ 521.0248, found 521.0255.

*2-Chloro-5-(5-(6-chloro-3-(ethylsulfonyl)pyridin-2-yl)-1,2,4-oxadiazol-3-yl)-N-(2-fluorophenyl)benzamide* (***10i***): Yellow solid, yield 71.6%, m.p. 229–231 °C; ^1^H NMR (500 MHz, DMSO-*d*_6_) δ 10.75 (s, 1H), 8.61 (d, *J* = 8.5 Hz, 1H), 8.24 (d, *J* = 2.0 Hz, 1H), 8.23–8.18 (m, 2H), 7.85 (d, *J* = 8.5 Hz, 1H), 7.78–7.71 (m, 2H), 7.22 (t, *J* = 9.0 Hz, 2H), 3.78 (q, *J* = 7.0 Hz, 2H), 1.26 (t, *J* = 7.5 Hz, 3H); ^13^C NMR (500 MHz, DMSO-*d*_6_) ^13^C NMR (126 MHz, DMSO-*d*_6_) δ 171.62, 166.79, 163.71, 157.51, 154.63, 142.94, 142.20, 137.61, 135.60, 134.97, 133.90, 131.18, 129.58, 128.78, 127.37, 124.27, 121.60, 115.43 (d, *J* = 88.0 Hz), 50.37, 6.62; HRMS calcd for C_22_H_16_Cl_2_FN_4_O_4_S [M + H]^+^ 521.0248, found 521.0251.

*2-Chloro-5-(5-(6-chloro-3-(ethylsulfonyl)pyridin-2-yl)-1,2,4-oxadiazol-3-yl)-N-(2,6-difluorophenyl)benzamide* (***10j***): Brown solid, yield 72.8%, m.p. 237–239 °C; ^1^H NMR (500 MHz, DMSO-*d*_6_) δ 10.56 (s, 1H), 8.61 (d, *J* = 8.5 Hz, 1H), 8.25 (m, 1H), 8.21 (d, *J* = 8.5 Hz, 2H), 7.89–7.82 (m, 2H), 7.42–7.34 (m, 1H), 7.19–7.12 (m, 1H), 3.78 (q, *J* = 7.5 Hz, 2H), 1.27 (t, *J* = 7.5 Hz, 3H); ^13^C NMR (500 MHz, DMSO-*d*_6_) δ 171.63, 166.79, 164.35, 154.64, 142.94, 142.18, 137.08, 135.61, 133.97, 131.19, 129.68, 128.79, 127.52, 127.17 (d, *J* = 7.5 Hz), 124.17, 111.40 (d, *J* = 12.5 Hz), 111.22, 104.60, 104.40 (d, *J* = 9.0 Hz), 104.19, 50.36, 6.58; HRMS calcd for C_22_H_15_Cl_2_F_2_N_4_O_4_S [M + H]^+^ 539.0154, found 539.0159.

*2-Chloro-5-(5-(6-chloro-3-(ethylsulfonyl)pyridin-2-yl)-1,2,4-oxadiazol-3-yl)-N-(2-chlorophenyl)benzamide* (***10k***): White solid, yield 82.0%, m.p. 209–212 °C; ^1^H NMR (500 MHz, DMSO-*d*_6_) δ 10.50 (s, 1H), 8.62 (d, *J* = 8.0 Hz, 1H), 8.34–8.14 (m, 3H), 7.93–7.82 (m, 1H), 7.73 (d, *J* = 7.0 Hz, 1H), 7.58 (d, *J* = 7.5 Hz, 1H), 7.48–7.38 (m, 1H), 7.36–7.29 (m, 1H), 3.86–3.73 (m, 2H), 1.28 (t, *J* = 7.0 Hz, 3H); ^13^C NMR (500 MHz, DMSO-*d*_6_) ^13^C NMR (126 MHz, DMSO-*d*_6_) δ 171.62, 166.79, 164.38, 154.66, 142.94, 142.18, 135.58, 134.07, 134.00, 131.23, 129.68, 129.63, 128.79, 128.77, 127.87, 127.84, 127.73, 127.59, 127.48, 124.15, 50.38, 6.61; HRMS calcd for C_22_H_16_Cl_3_N_4_O_4_S [M + H]^+^ 536.9952, found 536.9958.

*2-Chloro-5-(5-(6-chloro-3-(ethylsulfonyl)pyridin-2-yl)-1,2,4-oxadiazol-3-yl)-N-(2-chlorophenyl)benzamide* (***10l***): Yellow solid, yield 85.4%, m.p. 217–218 °C; ^1^H NMR (500 MHz, DMSO-*d*_6_) δ 10.91 (s, 1H), 8.62 (d, *J* = 8.5 Hz, 1H), 8.31–8.17 (m, 3H), 7.97–7.84 (m, 2H), 7.60 (d, *J* = 7.5 Hz, 1H), 7.42 (t, *J* = 8.0 Hz, 1H), 7.21 (d, *J* = 7.5 Hz, 1H), 3.79 (q, *J* = 7.0 Hz, 2H), 1.26 (t, *J* = 7.0 Hz, 3H); ^13^C NMR (126 MHz, DMSO-*d*_6_) ^13^C NMR (126 MHz, DMSO-*d*_6_) δ 171.63, 166.77, 164.09, 154.64, 142.93, 142.18, 139.97, 137.32, 135.58, 133.87, 133.15, 131.23, 130.58, 129.75, 128.79, 127.40, 124.30, 123.85, 119.19, 118.15, 50.37, 6.62; HRMS calcd for C_22_H_16_Cl_3_N_4_O_4_S [M + H]^+^ 536.9952, found 536.9959.

*2-Chloro-5-(5-(6-chloro-3-(ethylsulfonyl)pyridin-2-yl)-1,2,4-oxadiazol-3-yl)-N-(2-chlorophenyl)benzamide* (***10m***): White solid, yield 84.3%, m.p. 233–236 °C; ^1^H NMR (500 MHz, DMSO-*d*_6_) ^13^C NMR (126 MHz, DMSO-*d*_6_) δ 171.63, 166.77, 164.09, 154.64, 142.93, 142.18, 139.97, 137.32, 135.58, 133.87, 133.15, 131.23, 130.58, 129.75, 128.79, 127.40, 124.30, 123.85, 119.19, 118.15, 50.37, 6.62; ^13^C NMR (126 MHz, DMSO-*d*_6_) ^13^C NMR (126 MHz, DMSO-*d*_6_) δ 171.62, 166.78, 163.88, 154.64, 142.94, 142.19, 137.53, 137.47, 135.59, 133.88, 131.20, 129.66, 128.78, 128.75, 127.74, 127.39, 124.28, 121.29, 50.37, 6.62; HRMS calcd for C_22_H_16_Cl_3_N_4_O_4_S [M + H]^+^ 536.9952, found 521.0251.

*N-(4-Bromophenyl)-2-chloro-5-(5-(6-chloro-3-(ethylsulfonyl)pyridin-2-yl)-1,2,4-oxadiazol-3-yl)benzamide* (***10n***): Yellow solid, yield 78.3%, m.p. 263–265 °C; ^1^H NMR (500 MHz, DMSO-*d*_6_) δ 10.83 (s, 1H), 8.61 (d, *J* = 8.5 Hz, 1H), 8.33–8.11 (m, 3H), 7.85 (d, *J* = 8.0 Hz, 1H), 7.71 (d, *J* = 8.5 Hz, 2H), 7.56 (d, *J* = 8.5 Hz, 2H), 3.77 (q, *J* = 7.5 Hz, 2H), 1.26 (t, *J* = 7.5 Hz, 3H); ^13^C NMR (126 MHz, DMSO-*d*_6_) ^13^C NMR (126 MHz, DMSO-*d*_6_) δ 171.61, 166.77, 163.91, 154.65, 142.93, 142.18, 137.93, 137.44, 135.58, 133.87, 131.66, 131.21, 129.67, 128.78, 127.38, 124.27, 121.67, 115.81, 50.38, 6.61; HRMS calcd for C_22_H_16_BrCl_2_N_4_O_4_S [M + H]^+^ 580.9447, found 580.9449.

### 3.3. Biological Activity and Toxicity Determination

The insecticidal and fungicidal activities were investigated in the National Pesticide Engineering Research Centre, Nankai University, according to references [49,50], and the results of the activity test are shown in Table 1.

Through acute exposure, we assessed the toxicity of compounds **9f** and **10f** on zebrafish embryo. According to the preliminary exposure experiments, a series of gradient concentrations of compounds **9f** and **10f** were set on the basis of mortality rates in the range of 10–95%. LC_50_ values for zebrafish embryos exposed to compound **9f** or **10f** from 6 to 96 hpf: control (0 mg/L of **9f**), 1, 5, 10 mg/L of **9f**; control (0 mg/L of **10f**), 5, 10 mg/L of **10f**. The LC_50_ (median lethal concentration) values were computed by the Boltzmann equation [51,52]. The observational indexes included 96 hpf mortality rate, 72 hpf hatching rate, and 96 hpf malformation rate.

## 4. Conclusions

In conclusion, a series of novel benzamides containing 1,2,4-oxadiazole moiety were designed by bioisosterism and were synthesized easily via thioetherification, cyclization, aminolysis, and oxidation reactions. Their structures were confirmed by ^1^H NMR, ^13^C NMR, and HRMS. The bioassay results showed that some of the title compounds displayed excellent fungicidal activities against *Botrytis cinereal* at 100 mg/L. For example, **10a** (84.4%), **10d** (83.6%), **10e** (83.3%), **10f** (83.1%), **10i** (83.3%), and **10l** (83.6%) were better than the control fungicide pyraclostrobin (81.4%). In addition, the acute toxicity of **10f** to zebrafish embryo was 20.58 mg/L, which was classified as a low toxicity compound. Therefore, these compounds could potentially be the lead compounds for further study.

## Data Availability

Samples of the compounds are not available from the authors.

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
