# Peer review of "1,2,4-Oxadiazole-Based Bio-Isosteres of Benzamides: Synthesis, Biological Activity and Toxicity to Zebrafish Embryo"

_ijms, 2021, doi:10.3390/ijms22052367_

Round 1

Reviewer 1 Report

Current manuscript “Novel Benzamides Cantaining 1,2,4-Oxadiazole Moiety: Syn3 thesis, Biological Activity and Toxicity to Zebrafish Embryo” deals with the design and development of Broflanilide based 1,2,4-Oxadiazole possessing heterocyclic compounds. Moreover, these compounds were further evaluated for their antifungal activities against four different types of fungi. This paper expands the SAR study of Broflanilide, an insecticide development by Mitsui Chemicals and BASF pharmaceutical. While the activity optimization efforts provided only marginal improvement in fungicidal potency when compared with controls (Chlorantraniliprole and Pyraclostrobin) used in this study, key SAR conclusions (all three parts of the scaffold investigated are crucial for activity) are consistent with the set of compounds prepared herein. These findings are important and helpful to understand fungicidal activities of these series of compounds. The field will also benefit from the additional acute toxicity to zebrafish embryo described herein. Following points need to be addressed before considering ACCPETANCE.

  1. Scheme 1: authors should provide detailed information on ring substitution “R” instead of “etc”
  2. Scheme 2: Change Figure legend with “Synthesis of compound 3 via selective chloride substitution or thio-etherification” and provide reagents and conditions for this transformation separately.
  3. Author should add following references to support bio-isosteres applications in Drug discovery and development field:
  • Kohara, Y.; Kubo, K.; Imamiya, E.; Wada, T.; Inada, Y.;Naka, T. J. Med. Chem.1996,39, 5228.

(b) Tagad HD, Hamada Y, Nguyen JT, Hamada T, Abdel-Rahman H, Yamani A, Nagamine A, Ikari H, Igawa N, Hidaka K, Sohma Y, Kimura T, Kiso Y. Design of pentapeptidic BACE1 inhibitors with carboxylic acid bioisosteres at P1' and P4 positions. Bioorg Med Chem. 2010 May 1;18(9):3175-86.

  1. For Insecticidal and fungicidal activity authors used single concentration 500 mg/L and 100 mg/L respectively. It will be very informative if authors can provide a concentration dependent activity for these compounds (at least provide EC50 for most potent analog).
  2. A structure activity relationship study needs to be explained appropriately with detailed correlation between biological activities and structural modifications.
  3. All the compound reported in the manuscript possesses 1,2,4-Oxadiazole moiety in their core structure, so it will be very important to provide details on overall effect of “R” group substitution on the biological activity.
  4. As this series of compounds were based on Broflanilide, Authors should provide comparative biological activity of this compound under similar assay conditions.
  5. Overall manuscript is well written though requires few grammatical corrections.

Reviewer 2 Report

Sen Yang et al. wrote an interesting article on the development of benzamides containing 1,2,4-oxadiazole moiety. Their compounds showed good antifungal activity, in some cases the antifungal activity against Botrytis cinereal of the compounds is even better than that of the reference compound pyraclostrobin. Compound 10f containing ethylsulfonyl has low toxicity to zebrafish embryos and should be further explored. Errors Error: - Error in title: Cantaining should be Containing. - Reference 1 Missing. The article covers an important topic such as protection from fungi that cause a lot of damage in agriculture and is written in a fairly understandable way, there are still some typos in the article that the authors can correct with a spell checker. The article could be published after a minor revision.

Author Response

Dear reviewer:

Thank you very much for reviewing this manuscript. Your opinion is very pertinent and we have also revised it. This paper is a modification manual, please check.

Point 1: Error in title: Cantaining should be Containing.

Response 1: According to your suggestion, we checked and corrected the “cantaining” of this manuscript.

Point 2: Reference 1 Missing

  1. Response 2: According to your suggestion, We added the references “Verger-Philippe, P.J.P.; Boobis, A.R. Reevaluate pesticides for food security and safety, Science, 2013, 341, 717-718.; Wang, B.L.; Zhu, H.W.; Ma, Y. et al., Synthesis, insecticidal activities, and SAR studies of novel pyridylpyrazole acid derivatives based on amide bridge modification of anthranilic diamide insecticides, Agric. Food Chem. 2013, 61, 5483–5493.; Oerke, E.C. Crop losses to pests, J. Agric. Sci. 2006, 144, 31–43.”

Point 3: The article covers an important topic such as protection from fungi that cause a lot of damage in agriculture and is written in a fairly understandable way, there are still some typos in the article that the authors can correct with a spell checker.

Response 3: According to your suggestion, we checked and corrected the language and form of the manuscript.

We deeply appreciate your consideration of our manuscript, and we look forward to receiving the comments from you again.

Thank you and best regards.

Round 2

Reviewer 1 Report

The authors have corrected the manuscript according to the suggestions of both the reviewers. They have responded well on suggestions and corrections. However, the authors couldn’t present the concentration dependent activity of most potent analog of the series (EC50) as suggested in an earlier review. As for the evaluation of Insecticidal and Fungicidal activity, detail studies with IC50's are important and significant. Therefore, this manuscript can only be accepted for IJMS standards after modification.

(1) Author should provide EC50 value for a most potent compound of this study with concentration dependent activity.

(2) Title  "1,2,4-Oxadiazole Based Bio-isosteres of Benzamides: Synthesis, Biological Activity and Toxicity to Zebrafish Embryo."
